# Touching Emotions: How Touch Shapes Facial Emotional Processing Among Adolescents and Young Adults

**DOI:** 10.3390/ijerph22071112

**Published:** 2025-07-15

**Authors:** Letizia Della Longa, Teresa Farroni

**Affiliations:** Department of Developmental Psychology and Socialization, University of Padova, 35131 Padova, Italy; teresa.farroni@unipd.it

**Keywords:** adolescence, affective touch, emotion processing, facial expression, multisensory integration

## Abstract

Emotion recognition is an essential social ability that continues to develop across adolescence, a period of critical socio-emotional changes. In the present study, we examine how signals from different sensory modalities, specifically touch and facial expressions, are integrated into a holistic understanding of another’s feelings. Adolescents (*n* = 30) and young adults (*n* = 30) were presented with dynamic faces displaying either a positive (happy) or a negative (sad) expression. Crucially, facial expressions were anticipated by a tactile stimulation, either positive or negative. Across two experiments, we use different tactile primes, both in first-person experience (experiment 1) and in the vicarious experience of touch (experiment 2). We measured accuracy and reaction times to investigate whether tactile stimuli affect facial emotional processing. In both experiments, results indicate that adolescents were more sensitive than adults to the influence of tactile primes, suggesting that sensory cues modulate adolescents’ accuracy and velocity in evaluating emotion facial expression. The present findings offer valuable insights into how tactile experiences might shape and support emotional development and interpersonal social interactions.

## 1. Introduction

Emotion recognition is the process of making sense of an emotional experience, by detecting, evaluating, and interpreting our own and others’ feelings and it lays the foundations for the integration of different sensory information originating both from the external world and from within our internal body [1]. In daily social interactions, we do not experience emotions on the basis of a single sensory cue. Rather, emotions are conveyed and recognized through multiple sensory channels, which are often used together and combined to express complex feelings and anticipate others’ emotions, thoughts, and intentions [2]. Thus, emotion recognition is an dynamic process that occurs with integration of multisensory cues from the surrounding social environment [3]. Learning to recognize and appraise emotional information expressed in different ways and anticipate others’ feelings marks a critical socio-emotional skill during the sensitive period in which adolescents lay the groundwork for entering into the adulthood age [4,5]. During this period of life, when children begin to gradually detach from their parents to build more complex and independent relationships with peers, emotion recognition skills hold a central importance. However, how emotional processes develop and change across adolescence and how environmental factors may modulate socio-emotional developmental trajectories is still largely unknown [4]. Neurodevelopmental changes in brain function reciprocally interact with increasing social demands, relational changes, and different socio-affective contexts [5]. Thus, examining the interplay between multisensory experiences and recognition of dynamic emotional expressions is an opportunity to better understand how emotional processing develops during adolescence and whether it is supported by multisensory integration mechanisms.

### 1.1. Facial Emotion Recognition in a Developmental Perspective

In social communication, emotional cues, whether spontaneous or voluntary, are primarily transmitted though facial expressions. The capacity to accurately recognize and interpret emotional expressions from faces is an essential prerequisite for socio-emotional development and adaptive interpersonal interactions across a person’s lifespan [6]. From very early in life, faces represent extremely important social cues that capture newborn’s attention and involve specific neural responses [7,8,9]. Emotion recognition develops rapidly during infancy [10,11] and substantially improves during the preschool age [12,13,14,15]. However, less research has focused on whether and how the processing of emotional facial expression continues to develop and refine during adolescence. One possible limitation is related to the methods used to measure facial emotion recognition in developmental populations, which often include labeling tasks and emotion matching paradigms with static images of prototypical facial expressions. These tasks are prone to ceiling effects in performance [16], making them not suitable for studying later development of more sophisticated emotion recognition skills. Moreover, such tasks typically include unimodal information (face images) without considering the possible contribution of multisensory integration processes. The transitional period between childhood and adulthood is characterized by tremendous bodily and psychological changes alongside growing complexity of social relations and challenges [4,17,18]. The increased complexity of emotional experiences and interpersonal interactions requires more fine-tuned emotion recognition skills based on the integration of multiple sources of information to decode ambiguous facial expressions and subtle changes in emotions [5]. Indeed, when placed in front of multiple emotions [19] or morphed facial expressions [16], adolescents’ performance improved considerably with age, showing evidence for late developmental changes in the ability to recognize emotional expressions [16,19,20].

### 1.2. Neural Basis of Emotions: What Changes Across Adolescence

At the neural level, emotional face processing involves a network of brain areas, including the fusiform gyrus, prefrontal cortices, insula, and amygdala [21,22,23], which undergo important anatomic and functional changes throughout childhood and adolescence, possibly shaping the development of emotional processing [16,24,25,26]. Early in adolescence, subcortical circuitry dominates emotional responses, suggesting that adolescents might experience strong emotional reactivity, but slower development in brain structures linked with emotion regulation [27]. Thus, emotional vulnerability during adolescence, characterized by rapid fluctuation in emotions and moods, difficulties in recognizing and anticipating others’ emotion, and scarce regulation of their own emotional states, can be attributed to less capacity to regulate heightened emotional reactivity due to an unbalanced development of subcortical and frontal brain areas [28]. Brain development during the transition from childhood to adulthood is characterized by gradual changes in fronto-amygdala connectivity, with a proposed idea that these lead to more sustained prefrontal cortex engagement and effective regulation of subcortical limbic regions, progressively supporting emotional and behavioral self-regulation [25,29]. The temporal neurodynamics and the imbalance between subcortical limbic regions and prefrontal cortex during adolescence might lead to an increased risk for impulsive and emotionally driven suboptimal decisions and actions (e.g., high sensation seeking, emergence of anxiety and mood disorders; [30]). However, structural changes and functional reorganization of emotional brain circuits possibly reflects a sensitive period for adapting to the many new physical, affective, and social challenges [27,31].

From this perspective, adolescence can be redefined not only as a period of emotional vulnerability associated with risk behaviors, but it is also associated with greater brain plasticity, which represents an opportunity for learning and shaping brain networks through sensory and socio-affective experiences [32]. Thus, the implementation of new paradigms for the assessment of emotion recognition in a multisensory context provides a new lens to appreciate the rich interconnections across sensory and affective brain circuits that may regulate or intensify emotional processing. This area of research has important implications for detecting strengths and vulnerabilities during critical periods of development and for better understanding the potential role of providing adequate stimulation throughout different sensory channels that might support appropriate socio-emotional development. In particular, physical contact represents a very direct and intense channel of emotional exchanges, which typically supports intimacy and social bounding (i.e., love and sympathy; [2]), thus representing a strong affective cue that might be integrated with and modulate emotional information provided by other sensory modalities (i.e., facial expressions).

### 1.3. Neural Mechanisms of Affective Touch Across Development

The sense of touch provides an important point of connection and exchanges between our body and the external physical and social environment. Beyond haptic properties, touch has also been shown to carry a sense of social closeness and affiliation that promotes sharing of emotional experiences and creation or maintenance of social relationships [33,34]. This dimension of touch, often referred to as affective touch, relies on activation of a class of thin and unmyelinated fibers, C-tactile afferents, which optimally respond to specific properties of tactile stimulation resembling natural skin-to-skin contact (e.g., dynamic slow moving stimuli with skin temperature around 32 degrees; [35,36]). It has been hypothesized that the C-tactile system represents an evolutionary system involved in the rewarding value of physical contact and provides a neurophysiological foundation for social affiliation, by selecting and processing tactile information that is likely to carry socio-emotional relevance and by increasing motivation in seeking interpersonal physical contact between conspecifics [33,36]. Crucially, C-tactile afferents project to the posterior insula, a brain hub that supports the integration of external sensory cues (exteroception) with internal bodily signals (interoception), and it is tightly connected to the emotional parts of the brain [37,38].

The behavioral, physiological, and neural effects of affective touch indicate a crucial role of this sensory cue in building a bridge between the bodily self and the others, supporting and shaping social reward, affective bounding, learning, and emotion regulation throughout life, with special attention to sensitive periods in human development [39]. Although adolescence remains a underrepresented developmental age in the study of affective touch [40], initial evidence suggests that, at the subjective level, adolescents showed higher pleasantness ratings for affective touch compared to control stimulations, similar to those found in adults and possibly increasing from childhood to early adolescence [41,42,43]. More importantly, at the neural level, brain mechanisms associated with discriminative and affective dimensions of touch are largely established from childhood, while showing a continuing maturation of the secondary somatosensory cortex and right posterior superior sulcus [44]. Relative to adults, adolescents showed an enlarged neural reactivity to tactile stimuli, and in particular to affective touch, in various regions implicated in interoception and rewarding processing, including the bilateral posterior insula, inferior temporal gyrus, and striatum [45]. On the contrary, adolescents seem to recruit regulatory structures, such as the left anterior cingulate cortex, to a lesser extent than adults. Taken together, these results suggest that adolescents may be more susceptible to tactile stimuli, possibly influencing how they evaluate and approach social situations.

### 1.4. Affective Touch and Emotion Recognition

Because of its interoceptive quality, touch evokes immediate affective states, which may represent important embodied cues with the potential to modulate the way in which we perceive and interpret other social signals, such as emotional faces. Previous studies show that perception and neural processing of facial expressions are strongly influenced by the multisensory context, including verbal, visual, and auditory information [46]; however, much less attention has been paid to the role of tactile information in multisensory emotional processing. Initial evidence in this direction suggests that touch has the potential to shape the appraisal of emotional stimuli, modulating the valence and the salience of information from other sensory modalities. A study that combined vocal expressions and tactile stimulations showed that participants generally integrated both sources of information to evaluate the emotional valence [47]. Moreover, Pawling et al. (2017) found that neutral faces paired with affective touch were judged as more approachable than faces paired with non-affective touch, suggesting that affective touch can communicate a positive value, which can also a social stimuli it is associated with [48]. Finally, Ellingsen et al. (2014) found that affective touch can intensify the affective value of facial expressions, irrespectively of the valence. Indeed, participants judged smiling faces paired with touch as more friendly and attractive; on the contrary, angry faces were reported as less friendly and attractive [49]. These results suggest that touch has the potential to amplify the salience of emotional expression. Thus, it is still unclear whether tactile information acts as an emotional prime interacting with the valence of the facial expression rather than magnifying emotional salience in general.

### 1.5. Vicarious Experiences of Touch

Beyond direct physical contact, vicarious touch has been shown to modulate visual attention to faces and to promote emotion sensitivity, suggesting that observing tactile interactions among others has the potential to transmit socio-affective significance, resembling firsthand experiences of touch [50]. When presented with vicarious scenes of interpersonal contact, participants reported slow stroking as more pleasant and desirable than static or fast touch, indicating that their evaluations were attuned to CT-optimal touch [51]. Moreover, participants tended to report vicarious tactile sensations matching the location of touch observed in the videos [52]. At the neural level, vicarious touch has been shown to evoke overlapping neural representations in somatosensory areas with firsthand experience of being touched [53]. Notably, in the context of social interactions, activation of the posterior insula has been found in response to viewing videos of others being stroked, matching the neural blueprint of firsthand affective touch experiences [54]. Taken together, these results indicate that processing of vicarious touch is likely related to the human capacity for empathy toward others’ cognitive and emotional states. Thus, it is possible to hypothesize that perception of touch, whether felt or seen, may act as an emotionally significant cue that has the potential to modulate recognition and appraisal of others’ emotional expressions. Vicarious scenes of touch might be particular relevant in sensitive developmental periods. Adolescents’ attitudes and behaviors are influenced not only by direct sensory experience, but also by vicarious scenes of interpersonal contact [55].

### 1.6. The Present Study

In the present study, we investigate whether and how adolescents and young adults rely on sensory experiences to interpret facial emotional expressions. Across two experiments, we explore the effect of different types of touch, both experienced in first person (experiment 1) and observed between others (experiment 2), in modulating visual emotional processing, as reflected by response accuracy and speed of evaluation of dynamic emotional expressions. Participants were presented with 3 s videos of faces gradually displaying either a positive (from neutral to happy) or a negative (from neutral to sad) expression. Facial expressions were anticipated by a brief tactile stimulation, either positive or negative, so that in half of the trials tactile and visual information were congruent (both positive or both negative), while in other half they were incongruent (one positive and the other negative). In experiment 1, participants were presented with direct experiences of touch and we used both skin-to skin contact and object contact. In experiment 2, participants were presented with vicarious scenes of interpersonal touch (person-to-person) and object touch (person with object). We made two alternative hypotheses. On one hand, we may expect an interaction effect meaning that the tactile stimulation acts as an emotional prime that facilitates the processing of congruent facial expression based on emotional valence. On the other hand, it is also possible to hypothesize that positive touch, specifically targeting CT afferents, represents an interoceptive cue and an intrinsically affective reinforcement, thus promoting emotional processing in general. If this were the case, we may expect that positive touch would increase performance in recognizing facial expression irrespective to the emotional valence of the face, making participants more attuned to others’ emotions.

In a developmental perspective, we hypothesize that adolescents might be more influenced by tactile information when evaluating dynamic facial expressions compared to adults, resulting in a stronger interaction effect between valence of touch and valence of face. Considering the sensitive period of socio-emotional transition and functional reorganization of the brain that adolescents are facing [17], they may be more susceptible to affective cues from different sensory modalities to evaluate and interpret subtle changes of facial expressions, reflecting crucial abilities for adapting to one’s social environment.

Finally, we also considered possible differences of socially salient tactile stimulation (human touch vs. contact with an object). Although many studies on affective touch are based on tactile stimulations performed through objects (e.g., brush stroking), there is some evidence suggesting that direct skin-to-skin contact is processed differently from similar soft inanimate touch [56]. Indeed, stroking with a hand elicited larger responses than touch with a velvet stick in the contralateral primary and secondary somatosensory areas as well as in the posterior insula [56]. Based on these results, we expect that direct skin-to-skin touch might be more effective, compared to touch applied through inanimate objects, in understanding, and enhance the processing of, associated facial emotional expressions.

## 2. Materials and Methods

### 2.1. Participants

A total of 30 adolescents (16 females and 14 males) between the ages of 11 and 15 years (mean age 13.27 years) and 30 adults (16 females and 14 males) between the age of 19 and 33 years old (mean age 25.03 years) were included in the study. All participants were Caucasian. Parents of adolescents provided consent, while adult participants gave their own written consent after being informed about the study aims and procedure. The study was conducted in accordance with the ethical standards of the Declaration of Helsinki. The local ethical Committee of Psychological Research approved the study protocol (code 4725).

### 2.2. Stimuli and Procedure

At the start of the experimental session, participants were asked to comfortably sit on a chair in front of a table on which a laptop with touch screen was placed for presentation of visual stimuli. We used the software Labvanced (offline version) for presentation of visual stimuli and recording of participants’ responses across the experiment. It was also explained to participants that they would be touched on their arm during the experiment and if they were wearing a long-sleeved shirt, they were gently asked to roll up the sleeve to have easy access to the arm skin. At the beginning, we assessed individual pleasantness scores to different types of tactile stimuli (pre-test touch evaluation phase [57]). The tactile stimulations were performed on the participant’s dorsal forearm in a proximal-to-distal direction by a trained experimenter who sat beside the participant to have easy access to the participant’s exposed arm [58]. Touch gestures that communicate emotions are very rich and complex, composed by many characteristics (i.e., duration, location, intensity, velocity, texture, temperature). These different physical features of touch have an impact on the perception of emotions [59]. Pleasant emotions are typically associated with soft, slow, and dynamic touch, while negative emotions tend to be associated with short duration, high velocity, and high pressure [34]. In line with our previous studies [57,58,60,61], the stimuli were matched for the amount of sensory stimulation (i.e., force, contact area, and the stimulation rate), while they crucially differentiated for the socio-affective value conveyed by the source of touch (human skin-to skin contact vs. object contact) and the spatio-temporal dynamics (stroking vs. tapping). Specifically, we selected four different types of touch (see Figure 1):Positive social touch: gentle stroking applied by hand at velocity of approximately 3 cm/s;Negative social touch: rhythmic tapping with index, middle, and ring finger tips at rate of approximately one tap per second;Positive non-social touch: gentle stroking applied with a cosmetic brush (5 cm width) at velocity of approximately 3 cm/s;Negative non-social touch: rhythmic tapping with trident fork at rate of approximately one tap per second.

In order to assess whether these tactile stimulations were effective in evoking different sensations of pleasantness, participants were asked to rate the subjective pleasantness of each tactile stimulus on a visual analogue scale, e.g., [34]. To provide their evaluation, participants were ask to freely move their finger on a touch screen and select a point on a slider bar ranging from not pleasant at all (−10) to very pleasant (+10) [57].

After the tactile assessment, we began the experimental session consisting of two consecutive experiments, presented in counterbalanced order between participants. In both the experiments, the instructions and the task for the participant were the same, while we manipulated the type of tactile experience (direct touch vs. vicarious scenes of tactile interactions).

The experimental task consisted of evaluating the emotional valence of a face expression by pressing the key “X” if the face was positive (happy) and the key “N” if the face was negative (sad). Each facial expression was dynamic, starting from a neutral facial expression to assuming within 3 s the maximum intensity of either a positive or a negative emotion. Participants were instructed to provide a response as fast as possible and as accurately as possible. They were allowed to press any of the two bottoms at any time of the video presentation and the face would immediately disappear and a new trial would start. If participants did not press within the three seconds of video presentation, the face would remain on the screen displaying the maximum emotional expression until the response. Stimuli were selected from the Dynamic FACES database, a collection of morphed videos created by transitioning from a static neutral image to a target emotion based on the original FACES images [62]. Specifically, we selected Caucasian faces of young adults only (age range 18–30) of both female and male models.

In experiment 1, involving direct touch experience, we used the same stimulations presented in the pre-test evaluation phase. Each tactile stimulation lasted 3 s and it was immediately followed by presentation of a face stimulus. During the tactile stimulation, the display remained black and then a face appeared in the middle of the screen. We presented participants with 32 trials divided in two blocks based on the social dimension of touch (social block—human skin-to-skin contact; non-social block—object contact), while the valence of touch (positive vs. negative) was manipulated randomly within each block, making sure that half of the trials were congruent (both tactile and facial expression were either positive or negative) and half of the trials were incongruent (different valence between tactile and face expression).

In experiment 2, the experimental structure was the same as experiment 1, but instead of direct touch, we used videos of tactile interactions. In order to increase ecological validity and emotional salience, we decided to use videos representing interpersonal contact among bodies instead of simply representations of an arm being touched. Videos were selected from the socio-affective touch expression database [63]. Each video lasted 3 s and depicted a human–human interaction, either positive (i.e., different forms of gently stroking a person and hugging) or negative (i.e., punching and pushing a person on his/her arm), in the social block and a human–object interaction matched for the same movement properties in the non-social block (see Figure 1 for examples of videos). The videos have been created in the attempt to capture spontaneous, life-like touch expressions. Each actor pair consisted of one male and one female wearing either black or grey long-sleeved shirts and a turn performing as a touch initiator or touch receiver [63]. The object-based touch videos included situations where similar motions were required to interact with a particular object [63].

### 2.3. Data Analyses

As preliminary analyses, we compared subjective pleasantness score to tactile stimuli. Then, we ran main analyses on the emotion recognition task considering two dependent variables: accuracy and response time (RT). Before running the analyses, data were filtered excluding trials in which RTs were either shorter than 200 ms, indicating response anticipation, or longer than 6000 ms, indicating that the participant became distracted or took a break. A total of 40 trials of 3840 were excluded (1.04% of the total number of trials), resulting in 1907 valid observations for experiment 1 and 1893 for experiment 2. Only the trials in which participants provided a correct response (accuracy score = 1) were included in RT analyses. A total number of 162 trials were excluded as incorrect responses (accuracy = 0), resulting in 1810 valid observations for experiment 1 and 1828 for experiment 2.

All statistical analyses were performed using R, a software environment for statistical computing and graphics [64]. For each dependent variable, a set of different models were compared to select the model fitting our data best (model comparison approach). Specifically, we employed the Akaike Information Criterion (AIC) [65] and AIC weights [66] to evaluate the statistical evidence of each model and select the best fitting model uncovering the latent process behind the data, given the set of candidate models (lowest AIC value [67] and higher AIC weight). Then, we tested the predictors of interest included in the selected model, by using analysis of variance (type III Wald chi square test, “car” package [68]) and post hoc contrasts using the function “joint_tests” from the “emmeans” package [69]. Moreover we reported in Appendix A the summary of the best model using the function “sum” from the “jtools” package [70]. Generalized mixed-effect models allow for controlling for individual variability in repeated measure experimental designs [67]. Specifically, we used “glmer” function from the “lme4” package [71] and *p* values were also calculated using the “lmerTest” package [72]. We specify the most appropriate distribution for each dependent variable: Poisson distribution for subjective pleasantness (discrete number of possible outcomes); binomial distribution for accuracy (binary variable); and gamma distribution for RTs (positively skewed variable).

For subjective pleasantness, we considered a set of models as follows:Model 0 (null model): only accounted for individual variability;Model 1: included the main effect of Age group (adolescents vs. adults);Model 2: included the additive main effects of Age group and Valence of touch;Model 3: included the two-way interaction effect between Age group and Valence of touch;Model 4: included the two-way interaction effect between Age group and Valence of touch, with additive main effect of Social condition (human vs. object touch);Model 5: included the three-way interaction effect between Age group, Valence of touch, and Social condition;Model 6: included the three-way interaction effect between Age group, Valence of touch, and Social condition, with additive main effect of Gender.

For the emotion recognition task, we tested a set of predictors of interest for each dependent variable. These fixed effects included Age group (adolescents vs. adults), Face valence (positive vs. negative), Touch valence (positive vs. negative), Social condition (human vs. object touch), and Gender (female vs male). Additionally, all models included the random effect of participants to account for interpersonal variability. We considered the nine models as follows:Model 0 (null model): only accounted for individual variability;Model 1: included the main effect of Age group;Model 2: included the additive main effects of Age group and Face valence;Model 3: included the two-way interaction effect between Age group and Face valence;Model 4: included the two-way interaction effect between Age group and Face valence, with additive main effect of Touch valence;Model 5: included the three-way interaction effect between Age group, Face valence, and Touch valence;Model 6: included the three-way interaction effect between Age group, Face valence, and Touch valence, with additive main effect of Condition;Model 7: included the four-way interaction effects between Age group, Face valence, Touch valence, and Condition;Model 8: included the four-way interaction effects between Age group, Face valence, Touch valence, and Condition effect, with additive main effect of Gender.

## 3. Results

### 3.1. Pre-Test Subjective Preference for Touch

In order to ensure that our tactile manipulations were effectively differentiated according to valence, we analyzed the subjective scoring of pleasantness. We compared a set of seven nested mixed-effect models as described in Statistical Analyses section. According to the likelihood ratio test, the best fitting model was model 5 (AIC = 1276.2, ΔAIC = 4.45, wAIC = 0.35, χ^2^ = 10.45, *p* = 0.015; for details on model comparison, see Appendix A). The ANOVA on model 5 reveals that a main effect of Touch valence emerged (χ^2^ = 67.53, *p* < 0.001). indicating higher pleasantness scores for the positive tactile conditions (stroking compared to tapping). Although an interaction effect between Touch valence and Social condition emerged (χ^2^ = 4.38, *p* = 0.036), post hoc contrast indicated a significant effect of valence in both the Social (F = 59.18, *p* < 0.001) and Non-social tactile conditions (F = 33.40, *p* < 0.001; Figure 2; for details on the best model, ANOVA, and post hoc test, see Appendix A).

Moreover, to test for possible asymmetry in strength between positive versus negative tactile stimuli, we ran simple t-tests comparing each tactile condition to the neutral level (zero). Specifically, we performed eight t-tests and *p*-values were consequently adjusted for multiple comparisons using Bonferroni correction (*p* < 0.0062; see Figure 2). In both adolescent and adult groups, positive tactile conditions were reported as very pleasant and they significantly differentiated from the neutral level, whereas negative tactile conditions were reported as more neutral stimulations as they did not differentiate from the neutral level.

### 3.2. Experiment 1: Accuracy

To analyze participants’ accuracy in recognizing emotion facial expression, we compared a set of nine nested mixed-effect models as described in Statistical Analyses section. According to the likelihood ratio test, the best fitting model was model 5 (AIC = 549.4, ΔAIC = 6.73, wAIC = 0.34 χ^2^ = 12.73, *p* = 0.005; for details on model comparison see Appendix A). The ANOVA on model 5 reveals the main effect of Touch valence (χ^2^ = 4.42, *p* = 0.036) and an interaction effect between Face valence and Touch valence (χ^2^ = 10.36, *p* = 0.001), suggesting that participants are more accurate in recognizing the valence of facial expression when preceded by congruent tactile information. Importantly, a three-way interaction between Age group, Face valence, and Touch valence (χ^2^ = 9.01, *p* = 0.002) emerged. Post hoc analysis shows that the interaction between Face valence and Touch valence was significant in the adolescent group (F = 10.36, *p* = 0.001) but not in the adult group (F = 1.03, *p* = 0.310; Figure 3; for details on the best model, ANOVA, and post hoc test, see Appendix A).

### 3.3. Experiment 1: RT

To analyze participants’ speed in evaluating emotion expressions, we compared a set of models including the same factors used to analyze accuracy scores. According to the likelihood ratio test, the best fitting model was model 5 (AIC = 928.5, ΔAIC = 4.41, wAIC = 0.64, χ^2^ = 6.41, *p* = 0.011; for details on model comparison see Appendix A). The ANOVA on model 5 reveals the main effect of Age group (χ^2^ = 45.29, *p* < 0.001), indicating that adults (mean RT = 0.974 s, sd = 0.440) were faster than adolescents (mean RT = 1.910 s, sd = 0.837) in providing a response to facial expressions. Additionally, a main effect of Face valence emerged (χ^2^ = 9.86, *p* = 0.002), suggesting that, overall, participants were faster in responding to positive compared to negative facial expressions. This effect seems to be stronger in adult participants compared to adolescents, as indicated by the interaction effect between Age group and Face valence that emerged (χ^2^ = 6.42, *p* = 0.011) and by post hoc contrasts revealing that in both groups of participants, the effect of Face valence was significant, although with different effect size (adolescents: F = 9.86, *p* = 0.002; adults: F = 20.16, *p* < 0.001, Figure 4; for details on the best model, ANOVA, and post hoc test, see Appendix A).

### 3.4. Experiment 2: Accuracy

For the second experiment, we performed the same analyses presented in experiment 1. Considering accuracy scores, the best fitting model was model 5 (AIC = 502.7, ΔAIC = 5.43, wAIC = 0.55, χ^2^ = 11.43, *p* = 0.010; for details on model comparison, see Appendix A). The ANOVA on model 5 reveals the main effect of Face valence (χ^2^ = 7.02, *p* = 0.008) and an interaction effect between Face valence and Touch valence (χ^2^ = 9.06, *p* = 0.002), indicating that participants are more accurate in recognizing emotional expressions when preceded by a congruent vicarious scene of tactile interaction. More specifically, post hoc contrasts suggest this effect was especially driven by the negative tactile scene influencing face processing (F = 6.32, *p* = 0.012) compared to when the tactile scene was positive (F = 1.69, *p* = 0.193) (Figure 5; for details on the best model, ANOVA, and post hoc test, see Appendix A).

### 3.5. Experiment 2: RT

Considering the speed in evaluating emotion expressions, the best fitting model was model 6 (AIC = 779.9, ΔAIC = 3.89, wAIC = 0.80, χ^2^ = 5.89, *p* = 0.015; for details on model comparison, see Appendix A). The ANOVA on model 6 reveals the main effect of Age group valence (χ^2^ = 97.22, *p* < 0.001), indicating that adults (mean RT = 957 ms, sd = 455) were faster than adolescent (mean RT = 1865 ms, sd = 923) in providing a response and a main effect of Social condition (χ^2^ = 5.91, *p* = 0.015), indicated that, overall, participants were slower in the Social compared to the Non-social condition. Moreover, an interaction effect between Face valence and Touch valence emerged (χ^2^ = 11.28, *p* = 0.001), suggesting that participants are faster in recognizing emotional expressions when preceded by a congruent vicarious scene of tactile interaction. More specifically, post hoc contrasts suggest this effect was especially driven by the positive tactile scene influencing speed of face processing (F = 7.72, *p* = 0.006) compared to when the tactile scene was negative (F = 0.55, *p* = 0.458, Figure 6; for details on the best model, ANOVA, and post hoc test, see Appendix A).

## 4. Discussion

The present study shows evidence that tactile interactions might act as an emotional cue that modulates processing of emotional facial expressions, as indicated by interaction effects between the valence of touch and valence of face. Importantly, adolescents seem to be more influenced by tactile information than adults, as indicated by the different modulation of response accuracy in the two groups of participants, particularly in the context of direct tactile experiences (experiment 1). While previous studies highlight that perception of facial expressions is influenced by contextual information, focusing on visual and auditory stimuli presented together with faces [46], the potential role of touch in shaping emotional processes has received comparatively less attention. Touch is known to elicit affective states, suggesting that it may function as an embodied cue that modulates subsequent processing of emotional expressions. However, it remains unclear whether touch biases valence evaluations based on specific pleasant or unpleasant haptic parameters or exerts a more general effect by enhancing the salience of any emotional content. Both alternatives are supported by existing research. To explore these possibilities, we investigated the interaction between different tactile stimuli (positive and negative) and dynamic facial expressions assuming either a positive (happy) or a negative (sad) expression.

In the first experiment, participants experienced direct tactile interactions, either positive (caressing) or negative (tapping), applied by a human hand (social condition) or an object (non-social condition). Results show that adolescents, but not adults, were affected by tactile cues, displaying greater accuracy in recognizing emotional expressions when preceded by congruent tactile stimuli. Regarding response times, both adolescents and adults were faster in responding to positive compared to negative facial expressions, consistent with prior findings of an advantage in recognizing happy expressions across a person’s lifespan [73,74]. Overall, adults were faster than adolescents in evaluating dynamic facial expressions transitioning from neutral to peak emotional intensity. Age-related differences in processing speed, documented across various tasks, likely reflect both general and domain-specific components [75]. These differences tend to diminish with age, stabilizing in adulthood [75]. Although the developmental increase in processing speed may partly explain the age effect observed in response times, it is noteworthy that this study was not designed as a simple reaction task. In the present experiment, facial stimuli transitioned gradually from neutral to full emotional expression, resulting in a gradual acquisition of information: for adults, the mean response time was less than a second, while for adolescent it was almost two seconds. This significant difference suggests that adults are more sensitive to subtle emotional changes while adolescents rely more on additional information to interpret dynamic emotional expressions. Our results are in line with previous evidence suggesting late developmental changes in emotional expression recognition when stimuli are ambiguous (morphed faces; [16]).

Additionally, it is possible to speculate that contextual information might have a stronger impact on adolescents’ evaluations by integrating emotional cues from different sensory modalities to give sense to an emotional experience, showing a congruency effect between the valence of touch and the valence of emotional experience. Importantly, when presented with tactile cues and, in particular, soft touch, adolescents demonstrated greater activation in the posterior insula compared to adults, indicating increased interoceptive reactivity [45]. Thus, it is possible to speculate that adolescents may be more susceptible to contextual tactile cues when asked to evaluate an emotional facial expression. In everyday life, tactile exchanges provide important interoceptive signals, which crucially inform about others’ emotions and intentions. Adolescents might be particularly sensitive to these contextual cues, which, consequently, have the potential to shape how adolescents evaluate and approach social situations. Overall, our results suggest that tactile perception and facial emotion processing are interconnected, with developmental differences influencing their interplay. Rather than simply being a positive stimulus, touch is a complex embodied cue that has the potential to influence perception and interpretation of social signals, including emotional expressions.

Although we found the congruency effect only in the adolescent group, we cannot exclude that in more difficult conditions, adults might also be affected by contextual sensory information. Indeed, the accuracy scores of adults were on average 97%, suggesting a possible celling effect in the emotion evaluation task. Future studies could employ more subtle facial expressions to better investigate priming effects in adults.

Interestingly, contrary to expectations, socially salient tactile interactions (e.g., skin-to-skin contact) were not more effective in modulating emotional processing than touch applied with inanimate objects. Although previous evidence showed different neurophysiological responses to touch from a human hand versus an object [56], in an emotional context, it has been found that touch alters the event-related potentials to visual emotional stimuli independently of the source of stimulation. Schirmer et al. (2011) found similar effects whether the touch was applied by a friend, by a tactile device attributed to a friend, or by a tactile device attributed to a computer [76]. In line with this study, our results suggest that touch is an important sensory signal that influences emotional processing regardless of the social attribution. It is important to note that preliminary analyses on subjective pleasantness scores indicate that participants differentiated the valence of tactile stimulations based on the spatio-temporal dynamics (stroking vs, tapping) irrespective of the source of touch (skin-to-skin contact vs, object contact). Taken together, our findings suggest that the effects of touch depend more on subjective pleasantness than the source of stimulation, highlighting the importance of bottom-up sensory mechanisms in emotional priming.

In the second experiment, we included vicarious scenes of tactile interactions, taking a step further to explore whether the effects that emerged in the first-person experience of physical contact (experiment 1) could also be replicated with a simple observation of interpersonal touch between others. Results again show an interaction between the valence of touch and the valence of facial expression. Although the direction of the interaction effect is the same in both groups of participants, at the descriptive level, it was larger in adolescents than in adults (see Figure 5). Moreover, adults responded faster and showed an advantage for positive emotion expressions. Finally, an overall effect of social condition emerged, suggesting longer response time to emotional faces after observing human-to-human, rather than human-to-object, interactions. A speculative interpretation could be that observing interpersonal touch between others engages more attentional resources and high order interpretations of the scene, thereby requiring more time to disengage from the scene and evaluate the subsequent face.

In sum, results from experiment 1 and 2 provide a better understanding of cross-modal integration of emotional processing in adolescence, showing that both direct tactile experiences and vicarious scenes of interpersonal touch have the potential to influence adolescents’ evaluation of dynamic facial expressions. Considering that initially we made two alternative hypotheses—(I) presence of an interaction effect indicating that the touch acts as an emotional prime that facilitates the processing of congruent facial expression; and (II) increased performance in recognizing facial expression for positive touch targeting CT afferents as interoceptive cues promoting emotional processing in general— our results provide support for the first hypothesis, suggesting that touch is a complex embodied cue that has the potential to act as a primer and influence perception and interpretation of emotional expressions. Although differences in the stimuli prevent a direct comparison between the two experiments, at the descriptive level, we can observe that our results are in line with previous evidence indicating that direct and vicarious experiences of touch share some common neural activations and are both emotionally salient stimulations [50]. Indeed, considering accuracy scores in both experiments, interaction effects emerge between the emotional valence of the face and of the tactile stimulation with different modulation in the two groups of participants. In experiment 1, post hoc analysis shows that the interaction effect was significant only for adolescents but not for adults. In experiment 2, the three-way interaction was not significant, preventing further analyses separately in the two groups. Nevertheless, we can observe from descriptive plots (Figure 5) that the interaction effect seems to be stronger for adolescents compared to adults.

It is also important to mention some qualitative differences in the direction of the interaction effects in the two experiments. In experiment 1 (direct touch), adolescent participants were particularly sensitive to positive tactile cues, as suggested by a larger interaction effect between touch and the emotional valence of the face, when the tactile stimulation was positive, whereas the interaction effect seems more attenuated when the touch was negative. This observation may be related to the subjective perception of tactile stimuli. Indeed, tapping with a hand or fork might have been perceived as a neutral rather than a negative sensory cue, as suggested by mean pleasantness scores reported by participants (hand tapping: mean = 0.52 sd = 2.86; fork tapping: mean = −0.63 sd = 3.77, on a scale from −10 to +10; see Figure 1). Simple t-tests comparing each tactile condition to the neutral level (zero) further show that all negative tactile conditions were not significantly different from the neutral level. On the contrary, positive tactile conditions received very high pleasantness scores and they were significantly different from the neutral level, possibly suggesting an asymmetry in the strength of positive versus negative affective states evoked by the tactile stimuli. It is possible that increasing the intensity of negative touch could result in a stronger interaction effect; however, ethical issue should also be carefully considered when providing negative sensory cues that may be perceived as painful. On the contrary, in experiment 2 (vicarious touch), it is noticeable that participants, in particular adolescents, were less accurate in evaluating a positive face when preceded by a scene of negative tactile interaction. A possible interpretation is that high-level factors may have played a critical role in the evaluation of social interactions. Specifically, negative exchanges may have been perceived as very salient behaviors that should be generally avoided. Considering that it is likely that the negative stimuli considered involve a different emotional intensity in the two, the descriptive differences in the results seem to be related to the specific stimuli used rather than the tactile modality (direct vs. vicarious touch). Future investigation may require more similar stimuli (e.g., videos of arms touched in the same way as experienced in the direct condition) to make a direct comparison between first person and vicarious touch. Although only at the speculative level, these observations might suggest that the perceived intensity and salience of the stimulation plays an essential role in enhancing adolescents’ sensitivity to sensory cues in order to interpret dynamic facial expressions.

### Limits and Future Directions

First, we would like to acknowledge the relative small sample size of this study, which was mainly determined by the number of adolescents that we were able to recruit and agree to participate in the study. Based on the number of adolescent, we recruited a gender-matched group of adults. Given the complexity of the study design and the exploratory nature of the present research, along with the limited prior evidence to estimate expected effect sizes and determine an optimal sample size, our sample may not be large enough to detect additional differences between experimental conditions or age effects. Thus, we advise caution in interpreting the exploratory analysis, as future research, with larger inferential studies, will be necessary to confirm the findings presented here. Replication, meta-analyses, and multi-lab collaborations will be essential for addressing these issues in the broader process of knowledge acquisition, where each study contributes an important piece to the overall understanding of complex processes.

A second limit consists of presenting only two emotion expressions (happy and sad). In order to compared congruent and incongruent multisensory affective information, we selected two tactile stimuli and two facial expressions with opposite valence, while attempting to control for the level of stimuli intensity (for the positive condition, caressing touch and happy faces; for the negative condition, tapping touch and sad faces). Thus, the choice of employing sad faces instead of other negative and more arousing emotions, such as anger, was made to match as close as possible the relative low intensity of negative tactile stimuli (tapping). Please note that we decided to avoid painful stimulations to prevent any participants’ discomfort. Future studies may extend our initial results by including different emotion facial expressions and possibly different levels of arousal in order to explore not only the effect of congruent/incongruent valence between multisensory stimuli, but also how the intensity of the stimuli might modulate the interaction effect between conditions.

A final limit that we would like to mention consisted of employing only faces of young adults, in line with the majority of studies on facial emotion recognition in adolescents. This choice was made for the lack of an equal standardized database of morphed videos of adolescent faces and to ensure comparability between stimuli in the two groups of participants. However, we also recognize the importance of considering possible own-age bias. Only a few studies specifically investigated age-matched effects in emotion recognition across adolescence, showing inconsistent findings. Whereas one study provided support for increased accuracy for own-age faces [77], others studies found no evidence for own-age bias in adolescence [78,79]. Additionally, different neural responses to own-age compared to adult faces have been found in adolescents, while the accuracy performance was equal for both peers’ and adults’ faces [80]. Thus, future investigations should further examine this open issue and extend our initial findings by employing own-age faces to explore possible interactions effects between the valence of a peer’s emotional expressions and contextual sensory cues.

The present study paves the way for future lines of research, extending our findings with further investigations of the underlying neurophysiological mechanisms supporting multisensory integration and affective processing across development. Indeed, neural data may reinforce our initial evidence in favor of a priming effect of touch on facial expressions, in particular through the use of event-related potentials (ERPs). ERPs provide information about the temporal characteristics of processing facial expressions and represent a reliable marker of priming effects [81,82], which can be integrated with behavioral measures to explore whether tactile stimulation might facilitate processing of congruent facial expressions. It is also possible that tactile stimulation modulates the visual processing of faces, for example increasing the stimulus significance and focusing attention on specific facial features (i.e., eyes). The integration of an eye-tracking technique might extend the current investigation by providing additional information about possible attentional mechanisms supporting the role of touch in modulating processing of facial expressions.

## 5. Conclusions

In conclusion, learning to appraise emotional information and feelings with the buffer of sensory experiences represents a key milestone of the sensitive period in which adolescents prepare for adulthood. Adolescence can be viewed as a time when the emotional landscape is unmatched by that of childhood or adulthood, and a period of both emotional vulnerability and potential for development and redefinition of the interplay among emotions, behaviors, and social interactions [83]. This highlights the need to support and promote emotional processing. The present study underlines the importance of a multidimensional approach that acknowledges the ecological validity of dynamic stimuli and involves multiple sensory modalities. Our findings offer a new perspective on adolescent socio-affective development, pointing to the potential of tactile experiences as part of emotional processing. Results suggest that it is very important to take into consideration both first-person and vicarious experiences of physical contact that adolescents are exposed to, as these experiences play a significant role in shaping emotional processing, with possible cascading effects on engagement in effective social relationships across their lifespan. Thus, the present findings make a step further in understanding and promoting mental health and wellbeing among young adolescents and future adults, offering valuable insights into how tactile experiences might shape and support emotional development and interpersonal social interactions.

## Figures and Tables

**Figure 1 ijerph-22-01112-f001:**
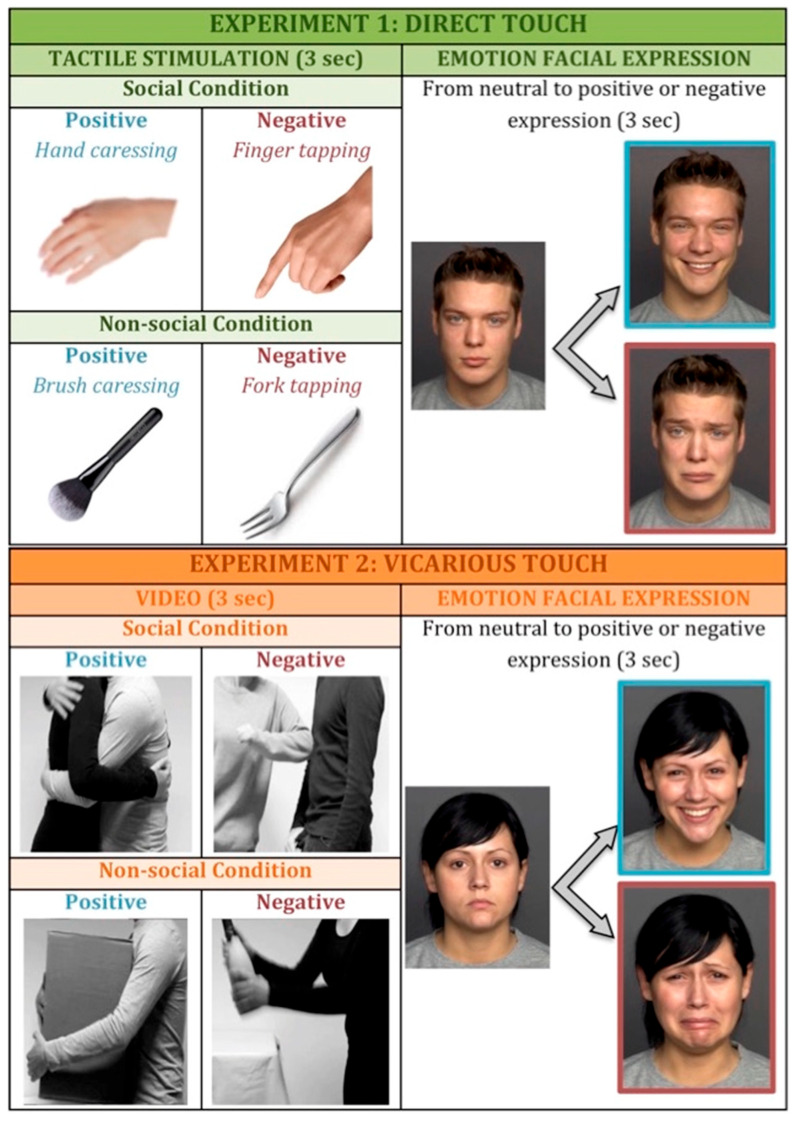
Experimental paradigm. In experiment 1 (**top panel**) participants were exposed to first-person experience of touch, while in experiment 2 (**below panel**) they watched videos of tactile interactions. In both experiments, we manipulated the valence of the stimulation (positive vs. negative) and the sociality (social vs. non-social condition). After the stimulation, participants were asked to respond as accurately and fast as possible to facial expression, pressing X if the face was assuming a positive expression and N if the face was assuming a negative expression.

**Figure 2 ijerph-22-01112-f002:**
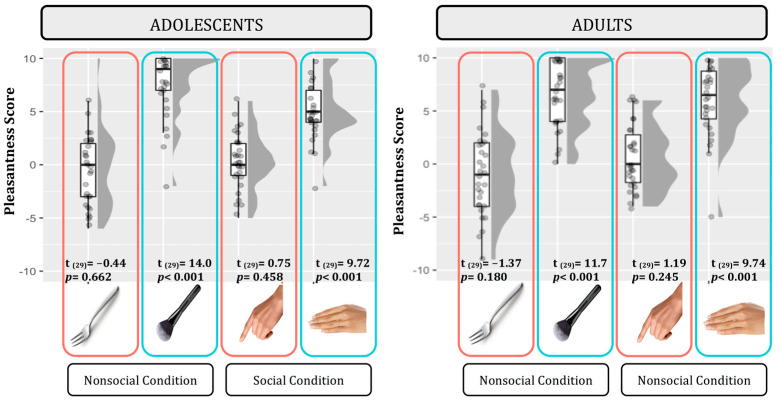
Pleasantness scores for each tactile stimulation in adolescents (**left panel**) and adults (**right panel**). Plots illustrate the central value (horizontal line), interquartile range (upper and lower limits of the boxes), individual scores (single points), and distribution of the data points (grey profile) for both positive (blue) and negative (red) tactile stimuli; *n* participants = 60; *n* observations = 240.

**Figure 3 ijerph-22-01112-f003:**
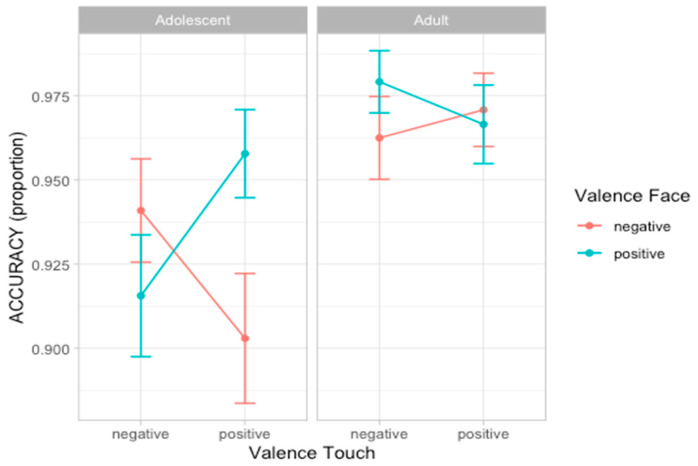
Plots of the accuracy scores showing the three-way-interaction effect between Age group, Face valence, and Touch valence. Dots represent the mean values and bars represent the standard errors. (*n* participants = 60; *n* observations = 1907).

**Figure 4 ijerph-22-01112-f004:**
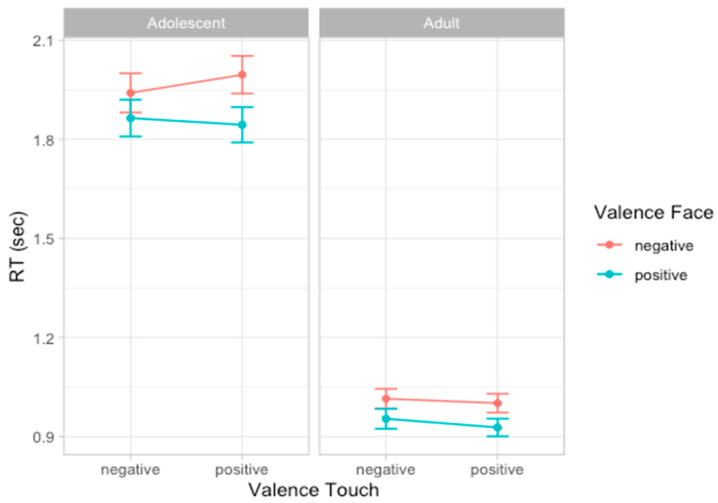
Plots of the effects predicting response time. Dots represent the mean values and bars represent the standard errors (*n* participants = 60; *n* observations = 1811).

**Figure 5 ijerph-22-01112-f005:**
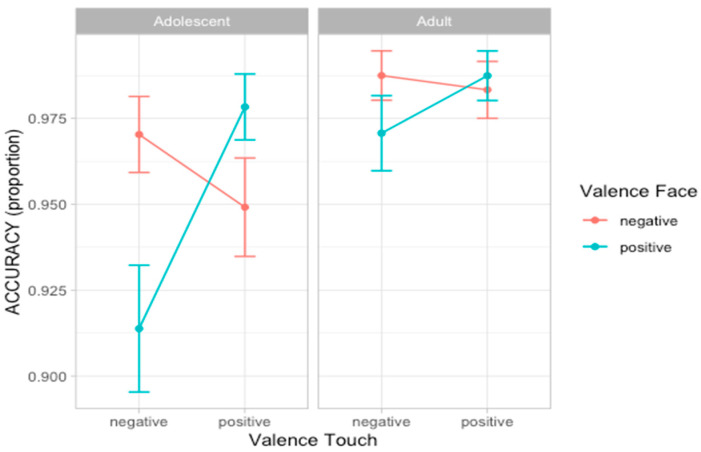
Plots of the effects predicting accuracy scores. Dots represent the mean values and bars represent the standard errors (*n* participants = 60; *n* observations = 1893).

**Figure 6 ijerph-22-01112-f006:**
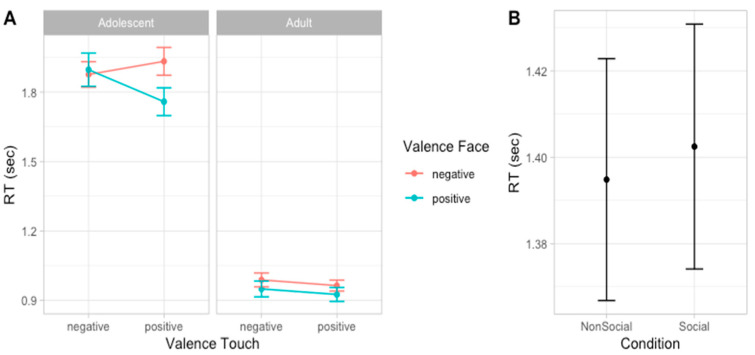
Plots of the effects predicting RTs. Dots represent the mean values and bars represent the standard errors. Left panel (**A**) shows the interaction effect between Face valence and Touch valence in adolescents and adults; right panel (**B**) shows the main effect of Social condition. (*n* participants = 60; *n* observations = 1832).

## Data Availability

The data and scripts for statistical analyses underlying this article are openly available at https://osf.io/rgc9a/ (accessed on 11 May 2025).

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
