# Peer review of "Touching Emotions: How Touch Shapes Facial Emotional Processing Among Adolescents and Young Adults"

_ijerph, 2025, doi:10.3390/ijerph22071112_

Round 1
Reviewer 1 Report (Previous Reviewer 4)
Comments and Suggestions for Authors
All of my comments have been addressed, except that the authors should include a citation to their previous work in the experimental section.
Author Response
All of my comments have been addressed, except that the authors should include a citation to their previous work in the experimental section. In addition, please appropriately reduce the repetition rate of the method section.
Response: We have now included citations of our previous works. For an example see: “In line with our previous studies [57,58,60,61], the stimuli were match for the amount of sensory stimulation (i.e. force, contact area and the stimulation rate), while they crucially differentiated for the socio-affective value conveyed by the source of touch (human skin-to skin contact vs object contact) and the spatio-temporal dynamics (stroking vs tapping).”
Moreover, we rephrased the method part to reduce the repetition rate. Please see the following paragraph with track-changes as an example:
“All statistical analyses were performed using R, a software environment for statistical computing and graphics [64]. For each dependent variable, a set of different models were compared to select the model fitting our data best (model comparison approach). Specifically, we employed the Akaike Information Criterion (AIC) [65] and AIC weights [66] to evaluate the statistical evidence of each model and select the best fitting model uncovering the latent process behind the data, given the set of candidate models (lowest AIC value [67] and higher AIC weight). Then we tested the predictors of interest included in the selected model, by using Analysis of Variance (type III Wald chi square test, “car” package [68]) and post-hoc contrasts using the function “joint_tests” from the “emmeans” package [69]). Moreover we reported in Supplementary Materials the summary of the best model using the function “sum” from the “jtools” package [70]. Generalized mixed-effect models allow to control for individual variability in repeated measure experimental designs [67]. Specifically we used “glmer” function from the “lme4” package [71] and p values were also calculated using the “lmerTest” package [72]. We specify the most appropriate distribution for each dependent variable: Poisson distribution for subjective pleasantness (discrete number of possible outcomes); binomial distribution for accuracy (binary variable); gamma distribution for RTs (positively skewed variable)”
Reviewer 2 Report (Previous Reviewer 3)
Comments and Suggestions for Authors
The new version of the paper, for what concerns reactions to Reviewer 3, can be endorsed.
Parallel improvements seem to show concerning the answers to the other reviewers.
Author Response
All comments have been addessed.
Reviewer 3 Report (New Reviewer)
Comments and Suggestions for Authors
see attached document

Round 2
Reviewer 1 Report (Previous Reviewer 4)
Comments and Suggestions for Authors
My comments have been addressed.
This manuscript is a resubmission of an earlier submission. The following is a list of the peer review reports and author responses from that submission.
Round 1
Reviewer 1 Report
Comments and Suggestions for Authors
This is a well written and clear manuscript describing an interesting set of experiments examining the role of tactile stimuli on the processing of visual emotional information conveyed by dynamic faces. The authors consider a first person experience of touch in Experiment 1 and a seen experience of touch in Experiment 2, analyzing the accuracy and reaction time to judge facial emotion in adolescents and young adults. Their main question is whether tactile information acts as an emotional prime, enhancing face processing based on the valence of the emotion conveyed by touch and by vision or if touch serves simply to amplify emotional processing more generally. The authors provide a detailed background of the behavioral and neural foundations of touch in emotional processing.
The experimental paradigm is well designed and clearly laid out. There are niceties of experimental design, such as a pre-test evaluation of touch stimuli. For positive touch they use stroking and for negative touch they use tapping. The source of the touch is human skin-to-skin contact or inanimate contact via a brush. Visual and tactile emotions are either congruent, matching, or incongruent, non-matching. A tactile stimulus is presented first for 3 seconds followed by a dynamic face, morphed from a neutral to an emotional face. Participants are tasked with evaluating the emotional valence conveyed by the face, positive (happy) or negative (sad).
MAJOR COMMENTS:
1. Can the authors provide a power analysis to show how sample size was determined.
2. The authors set out to show if tactile influences on emotional processing can be explained by priming versus a general amplification of emotional valence. Given that they find evidence for congruency effects, it seems that the first answer is the correct one, at least in adolescence. Why do the authors not state this more definitively in their discussion and conclusions? Can they elaborate a bit more at the end on what are the implications of them finding evidence for priming rather than a more general amplification?
3. The authors provide some important caveats to their results such highlighting that the negative stimuli (tapping) are not negative enough. Given that the authors conducted a pre-test evaluation of touch stimuli, can they provide this data to show the asymmetry in the strength of positive versus negative emotions and how this asymmetry might change with age?
MINOR COMMENTS:
Please see the list of typos or grammatical errors:
Line 150: “experience of being touch.” ïƒ “experience of being touched”
Line 212: tipically should be typically
Line 259: “instead that simply” -> “instead of simply using representations“
Figure 2 needs editing. The y axis labels and the word “nonsocial” need fixing. Also, there are “?” on the figure which maybe should be single points?
Line 425: missing “.” After word interaction.
Line 459: “salient cue” should be “salience cue”
Reviewer 2 Report
Comments and Suggestions for Authors
A high percentage of similarity with other research was detected, 31% according to the iThenticate report. After checking, it was found that the research uses literal quotations from other research and previous work by the authors already published in other journals. This means that in a significant percentage, the work has been constructed by copying and pasting from different sources, and the material is not original in its entirety. However, they extend previous research, so I suggest reconsidering their publication after an extensive revision, with a thorough original rewrite.
Other observations are some mixtures of reference standards (Pawling et al. (2017 line 128).
The research should be enriched with more socio-demographic data such as nationality, as emotions have a cultural component.
In the methodology, for experiment 1 and 2, the method of recording (technology) of the touching gesture that enables data collection is not explained clearly and in detail. The conditions under which the experiments were conducted are also not explained. This does not allow for replication of the experiment. There is also no clear justification for the use of the videos.
I suggest including a section on limitations and future lines of research.
I propose that the data and materials be uploaded to an open repository.
Reviewer 3 Report
Comments and Suggestions for Authors
The paper investigates how the combination of tactile stimuli affects emotion recognition from facial expressions in adolescents, as compared to adults. The results of two experiments show that adolescents are more sensitive than adults to the influence of tactile primes, both if directly touched and if observing someone touched, and, even, both if touched by a human or an inanimate object: touch cues modulate both accuracy and reaction times in facial expression interpretation.
A neat (although simple) paper, with grounded hypothesis, accurate methodology and results possibly useful for application in adolescents’ issues.
Minor comments
Some minor typos, e.g.,
“a emotional” instead of “an emotional”, line 512,
“although” or “whereas” followed by a comma, at lines 526, 547, 554
Concerning the use of inanimate objects’ touch in Experiment 1, one might observe that sometimes physical contact with objects may be reminiscent of contact with some persons, possibly affectively relevant.
Reviewer 4 Report
Comments and Suggestions for Authors
Reviewer Comments
The study investigates how signals from different sensory modalities like touch and facial expressions are integrated into a holistic understanding of another’s feelings among adolescents.
1. The experimental section in this manuscript closely resembles the previous work of the authors. Proper citation and differentiation between the two studies are necessary to clearly highlight the novelty of the current research. Authors may need to do extensive revision by paraphrasing cited work and extending the work.
2. Authors should restructure the article to include a separate introduction and related work section.
3. Overview of the study is missing. Authors should provide overview towards the end of the introduction.
4. In the abstract, please state concisely the background, motivation , method, results, and potential applications of findings of the study. The long sentences make it confusing. The abstract is made up of four long sentences.
5. Line 10 should read “ Emotion recognition is an essential social ability that continues …”
6. What are the contributions of this study?
7. What are the limitations of the study conducted?
Comments on the Quality of English Languagecould be improved.
